# Peroxymonosulfate Activation by Palladium(II) for Pollutants Degradation: A Study on Reaction Mechanism and Molecular Structural Characteristics

**DOI:** 10.3390/ijerph192013036

**Published:** 2022-10-11

**Authors:** Bowen Yang, Qiang Ma, Jiming Hao, Xiaojie Sun

**Affiliations:** 1Sichuan Provincial Engineering Research Center of City Solid Waste Energy and Buliding Materials Conversion & Utilization Technology, Chengdu University, Chengdu 610106, China; 2School of Architecture and Civil Engineering, Chengdu University, Chengdu 610106, China; 3State Key Joint Laboratory of Environment Simulation and Pollution Control, School of Environment, Tsinghua University, Beijing 100084, China; 4Guangxi Key Laboratory of Environmental Pollution Control Theory and Technology, Guilin University of Technology, Guilin 541004, China

**Keywords:** peroxymonosulfate (PMS), Pd(II), sulfate radical, BO_x_, E_HOMO_

## Abstract

Compared with certain transition metals (e.g., iron, cobalt, and manganese), noble metals are less frequently applied in peroxymonosulfate (PMS)-based advanced oxidation processes (AOPs). Palladium (Pd), as one of noble metals, has been reported to possess the possibility of both radical mechanisms and electron transfer mechanisms in a heterogeneous Pd/PMS system, however, data are still sparse on the homogeneous Pd/PMS system. Therefore, this work aims to explore the homogeneous reactivity of PMS by Pd(II) ions from the aspects of reaction parameters, radical or non-radical oxidation mechanisms, and the relationship between pollutants’ degradation rate and their molecular descriptors based on both experimental data and density functional theory (DFT) calculation results. As a result, the reaction mechanism of Pd(II)/PMS followed a radical-driven oxidation process, where sulfate radicals (SO_4_^•−^), rather than hydroxyl radicals (HO•), were the primary reactive oxidant species. BO_x_ and E_HOMO_ played significant roles in pollutant degradation during the Pd(II)/PMS system. It turned out that the bond’s stability and electron donation ability of the target compound was responsible for its degradation performance. This finding provides an insight into PMS activation by a noble metal, which has significant implications for scientific research and technical development.

## 1. Introduction

The fast development of industrialization can lead to not only the enhancement of the quality of human life, but a growing number of problems concerning environmental pollution, in particular wastewater as well. Thus, wastewater treatment has been considered one of the primary challenges over the past decades since it contains numerous pathogenic or disease-causing microorganisms and toxic compounds [1,2]. To this end, various technologies have been developed, such as adsorption [3], biological treatment [4], membrane separation [5], and advanced oxidation processes (AOPs) [6]. Unfortunately, the traditional wastewater treatment plants that are based on biological strategies (e.g., activated sludge and biofilm) are insufficient to successfully neutralize a line of emerging contaminants with high toxicity and refractory molecular structure. Accordingly, AOPs have been regarded as promising techniques for wastewater treatment owing to their easy operation and versatility, outstanding removal efficiency, minimal energy consumption, and simultaneous decontamination of multiple pollutants within a short time, even seconds [7,8,9].

AOPs that are based on sulfate radical (SO_4_^•−^), produced from peroxymonosulfate (PMS) and peroxydisulfate (PDS), have attracted considerable attention due to their high reactivity and selectivity in organic pollutants and complex environmental matrices [8,10]. Compared to conventional hydroxyl radicals (HO•), SO_4_^•−^ contains a series of merits: (i) SO_4_^•−^ has higher oxidation potential (2.5–3.1 V) than HO• (1.7–2.5 V) [11]; (ii) classical Fenton oxidation, which is commonly applied to generate HO•, works efficiently only in acidic conditions (pH ≤ 4.0) [12,13,14,15], while SO_4_^•−^ could react effectively with organic pollutants under a wide pH range of 2.0–12.0 [16,17,18]; and (iii) SO_4_^•−^ (30–40 µs) possesses a longer half-life period than HO• (<1 µs), which could enable sulfate radicals to have more stable mass transfer and react better with organic pollutants [19,20]. In this regard, several low-valent transition metals (i.e., Fe(II), Co(II), and Mn(II)) have been proven to be effective activators for wastewater treatment. For instance, Co(II) ion and cobalt oxides (i.e., Co_3_O_4_) could effectively transform PMS into SO_4_^•−^, thereby, the cobalt-induced PMS activation achieved outstanding treatment performance for wastewater involving 2,4-dichlorophenol (2,4-DCP), atrazine, naphthalene, bisphenol A (BPA), ciprofloxacin (CIP), chloramphenicol (CAP), and phenol [21,22,23,24,25].

In addition to transition metals, certain noble metals have recently been reported to be activators for PMS. Ahn et al. performed a comprehensive survey on 20 transition and noble metals. They found transition metals (i.e., Co, Cu, and Mo) could induce radical degradation mechanisms in the PMS system. In contrast, noble metals (i.e., Au, Ir, Pt, and Rh) followed electron transfer activation without involving the generation of radical species, such as HO• and SO_4_^•−^. Interestingly, palladium (Pd), as one of the noble metals, had the possibility of both radical mechanisms and electron transfer mechanisms [26]. By comparing Pd nanoparticles that were anchored on various supporters such as Al_2_O_3_, TiO_2_, SiO_2_, g-C_3_N_4_, C, and TiC, Feng et al., found that Pd-SiO_2_ obtained the highest reactivity of decomposing 1,4-dioxane in the PMS system. According to radical quenching tests, methanol was oxidized to formaldehyde (HCHO) by the whole Pd nanoparticles, which indicated that the degradation pathway followed a radical mechanism [27]. Furthermore, both the abovementioned works mainly focused on heterogeneous Pd, however, data are still sparse on the homogeneous Pd/PMS system.

Therefore, this study aims to further explore the homogeneous reactivity of PMS by Pd(II) ions, primarily focusing on (1) examining the effect of operating parameters such as the initial Pd(II) dosage, PMS amount, pH, and temperature; (2) exploring the main reactive oxygen species involving radicals (HO• and SO_4_^•−^) and non-radicals (^1^O_2_); (3) investigating the degradation performance of 16 pollutants containing pharmaceutical and personal care products (PPCPs), endocrine-disrupting chemicals (EDCs), dyes, as well as other emerging and traditional contaminants; and (4) revealing the relationship between the degradation rate and molecular structural characteristics based on the density functional theory (DFT) method. The findings from this work would provide an insight into PMS activation by noble metals, which has significant implications for scientific research and technical development.

## 2. Experimental

### 2.1. Chemicals and Materials

Phenol was regarded as a model contaminant for investigating the effect of operating parameters. A total of 16 organic pollutants were selected to systematically evaluate the reactivity of the Pd(II)/PMS system, namely carbamazepine (CBZ) and caffeine for PPCPs; bisphenol A (BPA) for EDCs, acid orange 74 (AO74) and rhodamine B (RB) for dyes; as well as phenol, 4-chlorophenol (4-CP), 2,4-dichlorophenol (DCP), 2,4,6-trichlorophenol (TCP), 4-nitrophenol (NP), 1,4-dioxane (1,4-D), 4-hydroxyphenylacetic acid (4-HBA), 4-nitrobenzoic acid (4-NBA), 4-nitroaniline (4-NA), benzoic acid (BA), and nitrobenzene (NB) for emerging and traditional contaminants. Oxone (2KHSO_5_^.^KHSO_4_^.^K_2_SO_4_) and Na_2_PdCl_4_ were used as the oxidant and activator, respectively. A complete list of the chemicals is shown in Appendix A.

### 2.2. Experimental Procedures

Unless otherwise specified, the degradation of different target pollutants (50 µM) was performed in a 50 mL reactor (50 mL wide-mouth glass bottle) with magnetic stirring of 800 rpm at room temperature under air-equilibrated conditions. The experimental suspensions typically consisted of a 0.25 mM PMS, and 25 µM Pd(II) ions activator. The initial solution pH was designed as 4.0 ± 0.1 based on containing 0.25 mM PMS. A total of 1 M of HClO_4_ or NaOH was used as a pH adjuster for investigating the effect of the initial pH. Sample aliquots were withdrawn from the reactor at the desired reaction time using a 1 mL syringe, filtered through a 0.45 μm PTFE filter (Millipore), then transferred into a 2 mL vial containing 20 µL MeOH (equal to ~0.5 M in 1 mL sample) to scavenge the possible residual oxidants. The brief experimental process is shown in Appendix A.

The concentrations of all the target pollutants were detected by high-performance liquid chromatography (HPLC, Agilent Infinity 1260, Santa Clara, CA, USA) that was equipped with a C-18 column and a UV/Vis detector. The mobile phase contained three tubes, namely H_3_PO_4_ solution (0.1%, *v/v*), pure acetonitrile (ACN), and pure MeOH. Formaldehyde (HCHO) that forms as a result of MeOH oxidation was quantitatively detected by HPLC after derivatization using 2 mM of 2,4-dinitrophenylhydrazine (DNPH) in ACN [28]. The details, such as solution ratio, wavelength, and flowrate, are presented in Appendix A. PMS was detected according to the method that was suggested by Liang et al., based on the generation of iodine (λ_max_ = 352 nm) from the oxidation of I^−1^ by PMS [29]. Bromate (BrO_3_^−^) was measured by an ion chromatograph (IC, Dionex DX120, Sunnyvale, CA, USA) that was equipped with a Dionex IonPac AS-14 and a conductivity detector.

### 2.3. Computation Details

The information on the molecular characters that were used in this work is listed in Appendix A. These descriptors have been successfully used in revealing the correlation between pollutant degradation performance and molecular parameters during other advanced oxidation processes (AOPs), e.g., Fenton oxidation, ozonation, and supercritical water oxidation [30,31,32,33,34]. The whole 17 molecular descriptors of the compounds were calculated by Material Studio 6.1 (Dmol3/GGA-BLYP/DNP(3.5) basis, Beijing, China) and Gaussian 09 (DFT B3LYP/6-311G level, Beijing, China). At first, the structure of the target compound was drawn and optimized through Gaussian 09 (DFT B3LYP/6-311G level). Thus, the exchange, correlation terms, and natural population analysis of atom charge were calculated by B3LYP function. Finally, certain molecular descriptors were achieved directly from the output files, such as dipole moment (μ), the total energy of a molecule in the B3LYP level (E_B3LYP_), the energy of the highest and lowest occupied molecular orbital (E_HOMO_ and E_LUMO_, respectively), the positive partial charge on a hydrogen atom (q(H)_x_), the negative or positive partial charge on a carbon atom (q(C)_n_ and q(C)_x_, respectively), and the positive partial charge of a hydrogen atom connected to a carbon atom (q(C-H)_n_ and q(C-H)_x_). As for bond order and Fukui indices, they were calculated via Material Studio 6.1 with DMol3 code, 6–311g (d,p) basis set and B3LYP function. A double numerical basis set with polarization functional was adopted at first. Afterwards, the density mixing and self-consistent field were set as 0.2 charge with 0.5 spin and 10^−6^ a.u., respectively. Then the results of bond order and Fukui indices were obtained from an outmol file, which included that the minimum and maximum number of chemical bonds between a pair of coterminous atoms (BO_n_ and BO_x_), the minimum or maximum value of Fukui indices by radical attack (F(0)_n_ and F(0)_x_), by nucleophilic attack (F(+)_n_ and F(+)_x_), and by electrophilic attack (F(−)_n_ and F(−)_x_). At last, the information on molecular descriptors was acquired from an outmol file.

## 3. Results and Discussion

### 3.1. Effect of Reaction Parameters

Several parameters, such as the Pd(II) dosage, PMS concentration, reaction temperature, and the initial pH, were examined for the degradation of phenol during the Pd(II)/PMS system (Figure 1). As expected, the degradation of phenol was largely enhanced with increasing Pd(II) dosage, PMS amount, and reaction temperature. In terms of Pd(II) dosage, the removal efficiency of phenol was 48% and 76% at 30 min for Pd(II) addition of 5 µM and 10 µM, respectively, while complete degradation of phenol was observed within 10 min and 2 min as Pd(II) dosage was enhanced to 25 µM and 50 µM, respectively (Figure 1a). For PMS concentration, merely 37% of phenol degradation was obtained at 0.1 mM PMS, and it increased to 76% as PMS was raised to 0.25 mM, then almost 100% was achieved within 5 min with promoting PMS to 0.5 mM and 1 mM (Figure 1b). As for the reaction temperature, the degradation efficiency of phenol was gradually improved to 60%, 76%, 89%, and 100% by augmenting the temperature from 4 °C to 65 °C (Figure 1c). Interestingly, the effect of pH showed a similar trend of phenol degradation in general (Figure 1d). This was attributed to the un-buffered solutions, where the presence of acidic Pd(II) and PMS stock solutions, as well as the release of H^+^ during the activation of PMS would lead to an unavoidable instant decline of pH (from the alkalescent or near-neutral to acidic condition). As displayed in Appendix A, the solution pH dropped to 2.9–4.1 at the end of the reaction. Besides, the control experiment regarding PMS or Pd(II) alone exhibited negligible phenol degradation (Appendix A), which elucidated that phenol removal by adsorption and direct PMS oxidation was marginal. As a result, the successful phenol degradation efficiency was mainly responsible for the activation of PMS by Pd(II). This was attributed to the decomposition of PMS by various Pd(II) dosages (Appendix A), therein, the more Pd(II) dosage that was added, the more PMS that was consumed after 30 min. Moreover, the homogenous Pd(II)/PMS system seems to display better phenol removal performance than the heterogeneous Pd-Al_2_O_3_/PMS system, where the removal efficiency of phenol was less than 70% [35].

### 3.2. Reaction Mechanism

Several alcohol-based scavengers such as methanol (MeOH) and *tert*-butyl alcohol (TBA) are commonly used in quenching experiments to identify whether radical (i.e., HO• and SO_4_^•−^) is the primary reactive oxidant species during PMS-AOPs [8,26,35]. In this regard, the degradation of phenol in the addition of MeOH and TBA was performed (Figure 2a), where the effect of TBA on phenol degradation was negligible. In contrast to that, an obvious quenching effect was found in the presence of MeOH. This was ascribed to SO_4_^•−^, rather than HO•, as the primary reactive oxidant during Pd(II)/PMS system, given that TBA is a specific HO• scavenger (k_HO•_ = 6 × 10^8^ M^−1^ s^−1^ versus k_SO4__•−_ = 7.6 × 10^5^ M^−1^ s^−1^) while MeOH can quench both HO• (k_HO•_ = 9.7 × 10^8^ M^−1^ s^−1^) and SO_4_^•−^ (k_SO4__•−_ = 2.5 × 10^7^ M^−1^ s^−1^) [11,36]. Additional evidence for the generation of SO_4_^•−^ was achieved by monitoring the oxidant production of formaldehyde (HCHO) from MeOH and bromate (BrO_3_^−^) from bromide (Br^−^) (Figure 2b). It showed a steep increase in both HCHO and BrO_3_^−^ concentration. The SO_4_^•−^-driven oxidation is undergo through facile one-electron oxidation of Br^-^ to Br• by SO_4_^•−^ (k = 3.5 × 10^7^ M^−1^ s^−1^) [36], then followed by oxidation to BrO_3_^−^ through intermediates (BrO^−^ and BrO_2_^−^) [37]. By contrast, the BA-to-4-HBA conversion failed (Figure 2b), which excluded that HO• played a significant role in the Pd(II)/PMS system, given that 4-HBA, as the main oxidant product of BA, was commonly detected during HO•-induced BA oxidation [38]. Analogous to the Pd(II)/PMS system, Co(II)/PMS, as a typical SO_4_^•−^-driven oxidation system [39], also displayed similar phenomena in the production of HCHO, BrO_3_^−^, and 4-HBA (Appendix A). Therefore, the reaction mechanism of Pd(II)/PMS probably followed SO_4_^•−^-driven oxidation.

To investigate whether ^1^O_2_ was involved in PMS activation by Pd(II), p-benzoquinone (p-BQ), furfuryl alcohol (FFA), azide ion (N_3_^−^), and L-histidine were introduced to the quenching experiment based on k = 9.8 × 10^8^ M^−1^ s^−1^ of (p-BQ) [40], 1.2 × 10^8^ M^−1^ s^−1^ (FFA), 1.0 × 10^9^ M^−1^ s^−1^ (N_3_^-^), and 1.5 × 10^8^ M^−1^ s^−1^ (L-histidine) for ^1^O_2_ [41,42]. As shown in Figure 3, phenol degradation was largely suppressed by the presence of p-BQ, FFA, N_3_^-^, and L-histidine. Whereas such inhibition was ascribed to the direct PMS depletion by these oxidant scavengers in their excess amounts (Appendix A), therein PMS was consumed around 98% for FFA, L-histidine, and N_3_^-^, and over 93% for p-BQ at the reaction time of 0.5 min, which was also in agreement with recent works [43,44]. To further determine the occurrence of ^1^O_2_ during the Pd(II)/PMS system, a control experiment was conducted in D_2_O solution according to D_2_O which could extend the lifetime of ^1^O_2_ by tenfold [45], thus the solvent exchange would promote the singlet oxygenation of pollutant [43]. In contrast, there was no enhancing effect on phenol degradation when H_2_O was replaced by D_2_O (Figure 3), which allowed us to discount the possible involvement of ^1^O_2_ as a reactive oxidant in the Pd(II)/PMS system.

### 3.3. Oxidative Degradation of Various Pollutants

In order to systematically evaluate the reactivity of PMS by Pd(II), sixteen kinds of organics on behalf of the emerging and traditional pollutants were introduced in our study. As displayed in Figure 4, various pollutants resulted in different degradation efficiencies, where nitrobenzene (NB) was removed by merely 38% at the reaction time of 30 min, while 4-CP, DCP, and TCP achieved 100% degradation efficiencies within 5 min. Such a discrepancy was attributed to the structural properties of pollutants. For instance, the electron-withdrawing groups (e.g., nitro and carboxylic groups in NB, 4-NBA, and BA) tend to be sluggish towards SO_4_^•−^, while the electron-donating groups (e.g., hydroxyl and chloride groups in phenol, 4-CP, DCP, and TCP) exhibited high susceptibility to SO_4_^•−^, which was analogous to the previous literature regarding sulfate radical-based AOPs [46]. In addition, the dyes like AO74 and RB acquired better degradation efficiencies, which contributed to their chromogenic group (i.e., diazo group) that could be easily attacked by radicals [47]. Similar results were also found in the recent work [35], where Ahn et al. investigated the heterogeneous Pd/PMS system using Pd-Al_2_O_3_ nanoparticles and observed the target compounds containing the electron-withdrawing groups (e.g., 4-NP, BA, and CBZ) were hardly removed, while the chemicals containing the electron-donating groups (e.g., phenol, BPA, 4-CP, and TCP) obtained better degradation performance.

### 3.4. Relationship between k and Molecular Descriptors

The quantum properties of these contaminants were calculated by DFT to explore the further the correlation between oxidative degradation and molecular structural characteristics (Appendix A). Furthermore, the degradation rate of pollutants was calculated by a pseudo-first-order kinetic model as follows:$$\ln\frac{{\left[C\right]}_{0}}{{\left[C\right]}_{t}}=kt$$
where [*C*]_0_ stands for the initial concentration of each compound (namely 50 µM), [*C*]*_t_* indicates its concentration at time t, *k* represents the degradation rate constants of each compound, and t is the reaction time. Accordingly, all the *k* values, their respective molecular descriptors, and the correlation coefficients of 16 target pollutants are listed in Appendix A. The correlation implied that q(H)_x_, q(C)_n_, F(−)_n_, F(−)_x_, F(+)_n_, F(+)_x_, F(0)_n_, and F(0)_x_ were positively correlated to *k*, while µ, E_B3LYP_, E_LUMO_, E_HOMO_, q(C)_x_, q(C-H)_n_, q(C-H)_x_, BO_n_, and BO_x_ were negatively correlated to *k*. The significance of molecular descriptors towards *k* on the basis of the absolute value followed the order that BO_x_ (−0.543) > E_HOMO_ (−0.509) > E_B3LYP_ (−0.467) > F(0)_x_ (0.447) > F(−)_n_ (0.408) > F(−)_x_ (0.377) > E_LUMO_ (−0.318) > F(0)_n_ (0.315) > F(+)_x_ (0.275) > q(C)_x_ (−0.265) > µ (−0.242) > q(H)_x_ (0.228) > F(+)_n_ (−0.210) > BO_n_ (−0.209) > q(C-H)_n_ (−0.203) > q(C)_n_ (−0.180) > q(C-H)_x_ (−0.101). It turned out that the most relative parameter was BO_x_, which greatly affected pollutant degradation in the Pd(II)/PMS system. Indeed, BO (bond order) reflects the number of the chemical bond between a pair of atoms, namely a higher value of BO could lead to more stability in the bond. In other words, the less the BO value was, the more easily it could be attacked, resulting in bond cleavage [48,49]. Besides, E_HOMO_, as another essential molecular parameter, is a measure of the electron donation ability of the target compound. This was in agreement with the finding of Section 3.3. Therein the compounds containing electron-withdrawing groups (e.g., nitro and carboxylic groups in NB, 4-NBA, and BA) presented relatively lower *k* values, while the chemicals possessing electron-donating groups (e.g., hydroxyl and chloride groups in phenol, 4-CP, DCP, and TCP) showed better degradation performance (Figure 4). Besides, compared with the relationship between *k* and molecular descriptors in other AOPs, such as the supercritical water oxidation (SCWO) process [33], where F(+)_n_, F(−)_n_, F(0)_n_, and E_HOMO_ achieved high correlation coefficients in the SCWO system, and followed an order that F(−)_n_ (−0.425) > F(+)_n_ (−0.417) > E_HOMO_ (0.415) > F(0)_n_ (−0.402), which was lower than the correlation coefficients achieved in this work (i.e., BO_x_ (−0.543) > E_HOMO_ (−0.509) > E_B3LYP_ (−0.467) > F(0)_x_ (0.447)). This indicates the correlation coefficients that were acquired in this work had more significant correlation than those in SCWO process.

## 4. Conclusions

The homogeneous PMS activation by Pd(II) has been found to exhibit highly effective degradation performance for pollutants. The degradation efficiency was enhanced with increasing Pd(II) dosage, PMS concentration, and reaction temperature, while the initial solution pH showed a similar trend of pollutant degradation in a wide un-buffered pH range of 3–9. Sulfate radical (SO_4_^•−^) was proposed as the dominant reactive oxidant species based on the quenching effect with excess MeOH and the high amount of Br^−^-to-BrO_3_^−^ conversion. In contrast, hydroxyl radicals (HO•) and singlet oxygen (^1^O_2_) played minor roles in the Pd(II)/PMS system. According to the correlation between the degradation rate (*k*) and molecular descriptors, BO_x_ and E_HOMO_ were the critical parameters for 16 target pollutants degradation during the Pd(II)/PMS system. In other words, the bond’s stability and electron donation ability of the target compound acted as an essential part of the pollutant degradation performance. In sum, the homogeneous Pd(II)/PMS system displayed an outstanding reactivity performance in a wide of pH and temperatures, therefore, a potential application of Pd(II)/PMS should be considered in practical wastewater treatment. Moreover, the findings from this work provide an insight into PMS activation by noble metals, which has significant implications for scientific research and technical development. Additionally, considering Pd is a kind of noble metal and its price is not economic, future work should focus on recycling the Pd(II) ions via introducing ligands, such as nitrilotriacetic acid (NTA), ethylenediamine tetraacetic acid (EDTA), oxalate acid, and citric acid.

## Figures and Tables

**Figure 1 ijerph-19-13036-f001:**
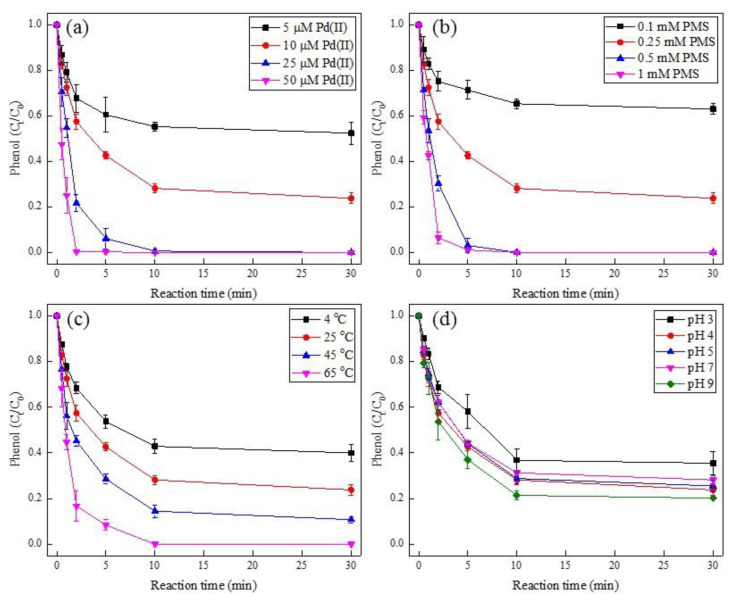
Effect of Pd(II) dosage (**a**), PMS concentration (**b**), reaction temperature (**c**), and initial pH (**d**) on the degradation of phenol. Condition: (**a**). [phenol] = 50 µM, [PMS] = 0.25 mM, pH_i_ = 4; (**b**). [phenol] = 50 µM, [Pd(II)] = 10 µM, pH_i_ = 4; (**c**). [phenol] = 50 µM, [PMS] = 0.25 mM, [Pd(II)] = 10 µM, pH_i_ = 4; (**d**). [phenol] = 50 µM, [PMS] = 0.25 mM, [Pd(II)] = 10 µM.

**Figure 2 ijerph-19-13036-f002:**
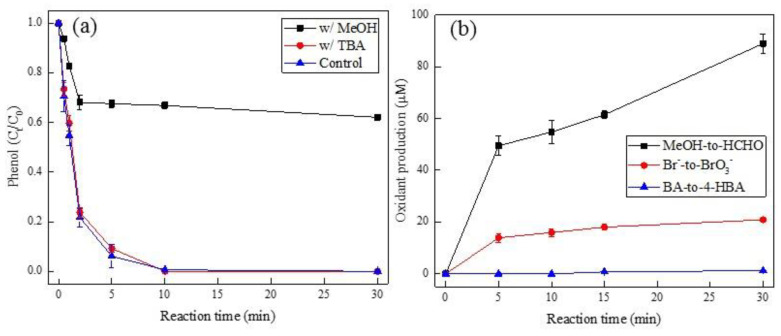
(**a**) The effect of MeOH and TBA on the degradation of phenol, and (**b**) oxidant production of HCHO, bromate, and 4-HBA in Pd(II)/PMS system. Condition: (**a**). [phenol] = 50 µM, [PMS] = 0.25 mM, [Pd(II)] = 25 µM, [MeOH] = [TBA] = 200 mM, pH_i_ = 4; (**b**). [MeOH] = 200 mM, [Br^−^] = 0.1 mM, [BA] = 10 mM, [PMS] = 0.25 mM, [Pd(II)] = 25 µM, pH_i_ = 4.

**Figure 3 ijerph-19-13036-f003:**
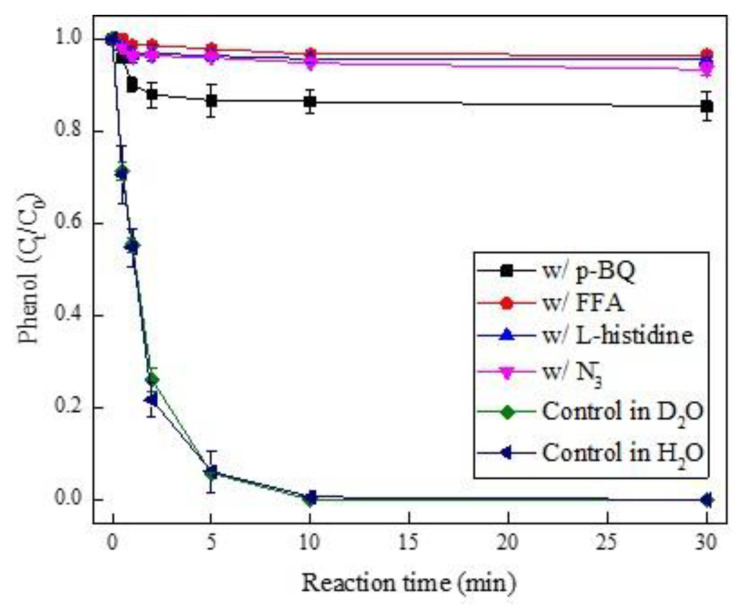
The effect of ^1^O_2_ scavengers and D_2_O solution on phenol degradation in the Pd(II)/PMS system. Condition: [phenol] = 50 µM, [PMS] = 0.25 mM, [Pd(II)] = 25 µM, [FFA] = [L-histidine] = [N_3_^−^] = 200 mM, [p-BQ] = 100 mM, pH_i_ = 4.

**Figure 4 ijerph-19-13036-f004:**
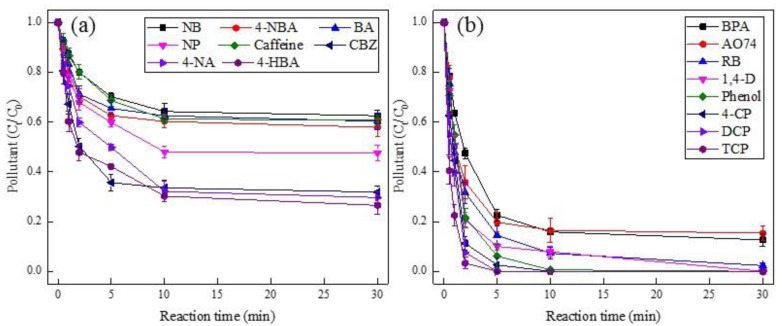
Degradation of various pollutants in Pd(II)/PMS system. Condition: [pollutant] = 50 µM, [PMS] = 0.25 mM, [Pd(II)] = 25 µM, pH_i_ = 4. (**a**) for the target compounds of NB, 4-NBA, BA, NP, Caffeine, CBZ, 4-NA, and 4-HBA; (**b**) for BPA, AO74, RB, 1,4-D, Phenol, 4-CP, DCP, and TCP.

## Data Availability

The data that support the findings of this study are available from the authors upon reasonable request.

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
