# Peer review of "Peroxymonosulfate Activation by Palladium(II) for Pollutants Degradation: A Study on Reaction Mechanism and Molecular Structural Characteristics"

_ijerph, 2022, doi:10.3390/ijerph192013036_

Round 1

Reviewer 1 Report

Review comments

This study investigated the Peroxymonosulfate activation by palladium (II) for pollutants degradation: a study on reaction mechanism and molecular structural characteristics. This manuscript needs careful modification. I feel that this work could only be possibly recommended for publication after minor revision, improvement.

Some comments are shown as following:

1.      The abstract is too general, please improve it. Also, include background statement into the abstract.

2.      What is the novelty of the study in comparison with the reference given? State the novelty of this work clearly at the end of introduction.

3.      Why the authors not compared the obtained results to other similar published studies?

4.      The experimental descriptions need to be more precise with sufficient details for reproduction.

5.      the work methodology can be improved by adding the following things:

·         study methodology flowchart

·         proposed structure of ANN

·         the different structure and flowchart of used models   

6.      Page 8, table 1, is overloaded, tried to make it clear

7.      The results part can be improved by more discussion of the results obtained.

8.      The authors should demonstrate that this is not an isolated case study. What are the general implications of the study? Comments on the general applicability and transferability of the results are needed.

Author Response

Response:

Thank you for your comments concerning our manuscript entitled “Peroxymonosulfate activation by palladium(II) for pollutants degradation: a study on reaction mechanism and molecular structural characteristics”. We have studied your comments carefully and have made correction which we hope meet with your approval. We are sending the revised manuscript according to the comments.

  1. We have modified and improved this part in Abstract, as follow:

Compared with transition metals (e.g., iron, cobalt, and manganese), noble metals are less frequently applied to peroxymonosulfate (PMS) based advanced oxidation processes (AOPs). Palladium (Pd), as one of noble metals, has been reported to possess the possibility of both radical mechanism and electron transfer mechanism in heterogeneous Pd/PMS system, however, data is still sparse on the homogeneous Pd/PMS system. Therefore, this work aims to explore the homogeneous reactivity of PMS by Pd(II) ions from the aspects of reaction parameters, radical or non-radical oxidation mechanism, and the relationship between pollutants’ degradation rate and their molecular descriptors based on both experimental data and density functional theory (DFT) calculation results. As a result, the reaction mechanism of Pd(II)/PMS followed a radical-driven oxidation process, where sulfate radical (SO4·-), rather than hydroxyl radical (HO•), was the primary reactive oxidant species. BOx and EHOMO played significant roles in pollutant degradation during Pd(II)/PMS system. It turned out that the bond’s stability and electron donation ability of the target compound was responsible for its degradation performance. This finding provides an insight into PMS activation by noble metal, which has significant implications for scientific research and technical development.

  1. What is the novelty of the study in comparison with the reference given? State the novelty of this work clearly at the end of introduction.
  2. Thank you for your suggestion, we have added the statement regarding the novelty of this work at the end of Introduction, as follows:

The findings from this work would provide an insight into PMS activation by noble metal, which has significant implications for scientific research and technical development.

  1. Why the authors not compared the obtained results to other similar published studies?
  2. Thank you for your advice. We have added some explaination refered to other similar published studies, as follows:

Besides, the homogenous Pd(II)/PMS system seems to display better phenol removal performance than the heterogeneous Pd-Al2O3/PMS system, where the removal efficiency of phenol was less than 70% [35] (see Line 170-172).

Similar results were also found in the recent work [35], where Ahn et al. investigated the heterogeneous Pd/PMS system using Pd-Al2O3 nanoparticles, and observed the target compounds containing the electron-withdrawing groups (e.g., 4-NP, BA, and CBZ) were hardly removed, while the chemicals containing the electron-donating groups (e.g., phe-nol, BPA, 4-CP, and TCP) obtained better degradation performance (see Line 240-244).

Besides, compared with the relationship between k and molecular descriptors in other AOPs, such as supercritical water oxidation (SCWO) process [33], where F(+)n, F(-)n, F(0)n, and EHOMO achieved high correlation coefficients in SCWO system, and followed an order that F(-)n (-0.425) > F(+)n (-0.417) > EHOMO (0.415) > F(0)n (-0.402), which was lower than the correlation coefficients achieved in this work (i.e., BOx (-0.543) > EHOMO (-0.509) > EB3LYP (-0.467) > F(0)x (0.447)) (see Line 277-282).

  1. The experimental descriptions need to be more precise with sufficient details for reproduction.
  2. Thank you for your suggestion. We have added more details on the experimental descriptions, as follows:

Unless otherwise specified, degradation of different target pollutants (50 µM) was performed in a 50 mL reactor (50 mL wide-mouth glass bottle) with magnetic stirring of 800 rpm at room temperature under air-equilibrated conditions. The experimental suspensions typically consisted of a 0.25 mM PMS, and 25 µM Pd(II) ions activator. The initial solution pH was designed as 4.0 ±0.1 based on containing 0.25 mM PMS. 1 M of HClO4 or NaOH was used as a pH adjuster for investigating the effect of initial pH. Sample aliquots were withdrawn from the reactor at desired reaction time using 1 mL syringe, filtered through a 0.45 μm PTFE filter (Millipore), then transferred into a 2 mL vial containing 20 µL MeOH (equal to ~0.5 M in 1 mL sample) to scavenge the possible residual oxidants (see Line 100-109).

  1. the work methodology can be improved by adding the following things:
  • study methodology flowchart
  • proposed structure of ANN
  • the different structure and flowchart of used models
  1. Thank you for your advice. We added a flowchart on the typically experimental process in this work, see Scheme S1 in Supplementary Materials.

  1. Page 8, table 1, is overloaded, tried to make it clear.
  2. We put the table in Supplementary Materials, namely Table S4.

  1. The results part can be improved by more discussion of the results obtained.
  2. Thank you for your suggestion, we have added more discussion on the results, as follows:

Besides, the homogenous Pd(II)/PMS system seems to display better phenol removal performance than the heterogeneous Pd-Al2O3/PMS system, where the removal efficiency of phenol was less than 70% [35] (see Line 170-172).

Similar results were also found in the recent work [35], where Ahn et al. investigated the heterogeneous Pd/PMS system using Pd-Al2O3 nanoparticles, and observed the target compounds containing the electron-withdrawing groups (e.g., 4-NP, BA, and CBZ) were hardly removed, while the chemicals containing the electron-donating groups (e.g., phe-nol, BPA, 4-CP, and TCP) obtained better degradation performance (see Line 240-244).

Besides, compared with the relationship between k and molecular descriptors in other AOPs, such as supercritical water oxidation (SCWO) process [33], where F(+)n, F(-)n, F(0)n, and EHOMO achieved high correlation coefficients in SCWO system, and followed an order that F(-)n (-0.425) > F(+)n (-0.417) > EHOMO (0.415) > F(0)n (-0.402), which was lower than the correlation coefficients achieved in this work (i.e., BOx (-0.543) > EHOMO (-0.509) > EB3LYP (-0.467) > F(0)x (0.447)) (see Line 277-282)

  1. The authors should demonstrate that this is not an isolated case study. What are the general implications of the study? Comments on the general applicability and transferability of the results are needed.

8.Thank you for your advice, we have added several comments on general applicability and transferability of this work, as follows:

In sum, the homogeneous Pd(II)/PMS system displayed an outstanding reactivity performance in a wide of pH and temperature, therefore, a potential application of Pd(II)/PMS should be considered in practical wastewater treatment. Moreover, the findings from this work provide an insight into PMS activation by noble metal, which has significant implications for scientific research and technical development (see Line 299-303).

Thank you again!

Reviewer 2 Report

The paper by Yang et al. presents an interesting topic on the reaction mechanism and molecular structural characteristics of PMS by Pd(II) for pollutant degradation. The article is well written and logically structured adding value to the wastewater treatment processes. The authors' findings are novel, however, I would like to request that the authors point out the novelty and originality of their work in a more precise way; perhaps in the last paragraph of the introduction. 

Author Response

Response:

Thank you for your comments concerning our manuscript entitled “Peroxymonosulfate activation by palladium(II) for pollutants degradation: a study on reaction mechanism and molecular structural characteristics”. We have studied your comments carefully and have made correction which we hope meet with your approval. We are sending the revised manuscript according to the comments.

We have added the statement regarding the novelty of this work at the end of Introduction and Conclusions, as follows:

The findings from this work would provide an insight into PMS activation by noble metal, which has significant implications for scientific research and technical development (Line 85-87).

In sum, the homogeneous Pd(II)/PMS system displayed an outstanding reactivity perfor-mance in a wide of pH and temperature, therefore, a potential application of Pd(II)/PMS should be considered in practical wastewater treatment. Moreover, the findings from this work provide an insight into PMS activation by noble metal, which has significant impli-cations for scientific research and technical development (Line 299-303).

Thank you again!

Reviewer 3 Report

The manuscript deals with the homogeneous reactivity of peroxymonosulfate (PMS) by palladium (Pd(II)) ions. One of the most important part of the manuscript is related to the correlation between oxidative degradation and molecular structural characteristics. However, just that part (i.e. Table 1) contains, in my opinion, significant shortcomings and my main objections are as follows:

1. The maximum (absolute) value of correlation coefficients is below 0.6. This indicates, according to the usual interpretation, that there is (only) weak correlation between the corresponding variables. Therefore, I believe that some other, relatively simple, but more sophisticated statistical methods should be additionally applied here. In my opinion, it would be prefer to establish a regression (or some other) multidimensional model between the degradation rate (k) and molecular descriptors ​​of 16 target pollutants as the independent (explanatory) variables.

2. The explanation of the results, i.e., the correlation coefficients (lines 260-265) must be more precise. In fact, their impact should be seen in their absolute values. For instance, mathematically speaking, value of BOx(-0.543) is smaller than EHOMO (-0.509), but the correlation impact of BOx is greater than EHOMO, etc.

Minor suggestions:

1- Line 250: The formula should be retype using the Equation Editor.

2- Line 255: Part of the sentence ".. were obtained based on the data from Figure 4." not adequate at all. Data can be obtained (only) by collecting them through some process of observation and measurement.

3- Table 1 and all Figures should be improved aesthetically, because their display is not the best visually.

Author Response

Response:

Thank you for your comments concerning our manuscript entitled “Peroxymonosulfate activation by palladium(II) for pollutants degradation: a study on reaction mechanism and molecular structural characteristics”. We have studied your comments carefully and have made correction which we hope meet with your approval. We are sending the revised manuscript according to the comments.

  1. Thank you for your suggestion. The highest correlation coefficient in this work is (lower than 0.6), which turned out there was weak correlation between the corresponding variables. However, compared with the relationship between k and molecular descriptors in other AOPs, such as supercritical water oxidation (SCWO) process [33], where F(+)n, F(-)n, F(0)n, and EHOMO achieved high correlation coefficients in SCWO system, and followed an order that F(-)n (-0.425) > F(+)n (-0.417) > EHOMO (0.415) > F(0)n (-0.402), which was lower than the correlation coefficients achieved in this work (i.e., BOx (-0.543) > EHOMO (-0.509) > EB3LYP (-0.467) > F(0)x (0.447)). This indicates the correlation coefficients acquired in this work had more significant correlation than those in SCWO process. Furthermore, quantitative structure-activity relationship (QSAR) model would be a preferable method to further explore the relationship between the degradation rate (k) and molecular descriptors. As a result, we will complete the QSAR model regarding Pd(II)/PMS system in our future study.

Reference:

[33] Yang, B.; Cheng, Z.; Tang, Q.; Shen, Z., Nitrogen transformation of 41 organic compounds during SCWO: A study on TN degradation rate, N-containing species distribution and molecular characteristics. Water Research 2018, 140, 167-180.

  1. The explanation of the results, i.e., the correlation coefficients (lines 260-265) must be more precise. In fact, their impact should be seen in their absolute values. For instance, mathematically speaking, value of BOx(-0.543) is smaller than EHOMO (-0.509), but the correlation impact of BOx is greater than EHOMO, etc.

  1. Thank you for your advice. We have modified this point, as follows:

The significance of molecular descriptors towards k on the basis of the absolute value followed the order that BOx (-0.543) > EHOMO (-0.509) > EB3LYP (-0.467) > F(0)x (0.447) > F(-)n (0.408) > F(-)x (0.377) > ELUMO (-0.318) > F(0)n (0.315) > F(+)x (0.275) > q(C)x (-0.265) > µ (-0.242) > q(H)x (0.228) > F(+)n (-0.210) > BOn (-0.209) > q(C-H)n (-0.203) > q(C)n (-0.180) > q(C-H)x (-0.101) (see Line 265-269).

Minor suggestions:

1- Line 250: The formula should be retype using the Equation Editor.

  1. We have modified this issue in Line 257, as follows:

 = kt

2- Line 255: Part of the sentence ".. were obtained based on the data from Figure 4." not adequate at all. Data can be obtained (only) by collecting them through some process of observation and measurement.

  1. Thank you for your suggestion. We have deleted this sentence.

3- Table 1 and all Figures should be improved aesthetically, because their display is not the best visually.

  1. Considering Table 1 is overloaded, we put the table in Supplementary Materials, namely Table S4. Furthermore, we also redrew the Figures according to your suggestion.

Thank you again!

Round 2

Reviewer 3 Report

In this revised version of the manuscript, the authors have fulfilled most of my objections. Perhaps Table that was in the previous version, with certain corrections, could have remained as it provided additional information about the results obtained. In any case, I have nothing against the work being accepted in this form, if, of course, the manuscript itself be visually and technically improved.